## PROCEEDINGS A

# Research

physical chemistry, crystal engineering, chemical engineering

calcium carbonate, crystal habit, Ostwald rule of stages, polymorphism, surface energy

**Author for correspondence:**
Eftychios Hadjittofis
e-mail: fc4eh@sheffield.ac.uk

# Exploring the role of crystal habit in the Ostwald rule of stages

Eftychios Hadjittofis[1,2], Silvia M. Vargas[3], James D. Litster[1] and Kyra L. Sedransk Campbell[1]

[1]Department of Chemical and Biological Engineering, University of Sheffield, Mappin Street, Sheffield S1 3JD, UK
[2]UCB Pharma SA, 1420 Braine l'Alleud, Belgium
[3]BP Exploration, 1330 Enclave Parkway, Houston, TX 77077, USA

EH, 0000-0001-8107-8368

The crystallization of calcium carbonate is shown to be dictated by the Ostwald rule of stages (ORS), for high relative initial supersaturations ($S_{CaCO_3} = [Ca^{2+}][CO_3^{2-}]/K_{SP, Calcite} > 2500$), under sweet (carbon dioxide saturated) and anoxic (oxygen depleted) solution conditions. Rhombohedral calcite crystals emerge after the sequential crystallization and dissolution of the metastable polymorphs: vaterite (snowflake-shaped) and aragonite (needle-shaped). However, the presence of certain cations, which can form trigonal carbonates (e.g. $Fe^{2+}$ and $Ni^{2+}$), in concentrations as low as 1.5 mM, triggers the emergence of calcite crystals, with a star-shaped crystal habit, first. These star-shaped crystals dissolve to yield needle-shaped aragonite crystals, which in turn dissolve to give the rhombohedral calcite crystals. The star-shaped crystals, formed at high $S_{CaCO_3}$, possess higher surface free energy (therefore higher apparent solubility) than their rhombohedral counterparts. This sequence of dissolution and recrystallization demonstrates that the ORS does not only drive the crystal towards its thermodynamically most stable polymorph but also towards its most stable crystal habit.

## 1. Introduction

In 1897 [1,2], the Nobel Laureate Wilhelm Ostwald wrote in relation to the crystallization of salts:

> When leaving a given state and in transforming to another state, the state which is sought out is not the thermodynamically stable one, but the state nearest in stability to the original state.

This statement, the backbone of the Ostwald rule of stages (ORS), describes the polymorphic transformations occurring during crystallization at high supersaturations ($S_{CaCO_3} = [Ca^{2+}] \cdot [CO_3^{2-}]/K_{sp'Calcite}$). It has been widely reported that the formation of the most thermodynamically stable polymorph is the outcome of a sequential process, where the pathway is comprised of crystallization and dissolution of metastable polymorphs [3–7]. All polymorphic forms should have the same nominal solubility; equal to the solubility of the most stable polymorph. Metastable, i.e. less thermodynamically stable, polymorphs can demonstrate a solubility higher than their nominal one. As such, apparent solubility is defined to describe this phenomenon: apparent solubility is the maximum solubility that can be observed for a certain polymorph, under discrete conditions. The existence of apparent solubility is a consequence of the lower energetic penalty required for the solvation of crystalline molecules at the solid–liquid interface for some metastable polymorphs as compared to counterparts. Therefore, it can be suggested that the discrepancy between the apparent solubilities of either two metastable polymorphs, or a metastable polymorph and the corresponding thermodynamically stable polymorph [8], is driving the transformations described by the ORS.

One could then query how the apparent solubility of two different crystals of the same polymorph, where one has the most stable crystal habit (i.e. a crystal habit which minimizes its surface free energy), would also have distinct apparent solubilities. If the hypothesis is extended, the crystal with the most stable habit should exhibit a lower apparent solubility as compared to crystals formed with less stable habits. This occurs because the crystals contain different facets, which will each contribute to the solvation energies for the crystal. This is akin to the different surface energy contributions from individual facets to the crystal as a whole [9,10]. Thus, a deviation from the most stable crystal habit will result in crystals with a metastable crystal habit, where the total solvation energy requirements will be higher. In a recent paper, the authors demonstrate this possibility in the context of two different crystal habits of calcite: the established rhombohedral one and a star-shaped one. Using a combination of thermogravimetric analysis, surface energy measurements and thermodynamic calculations, they demonstrated that the particles of a stable polymorph, with a morphology of sufficiently high surface energy, may be, in fact, metastable with respect to particles of a presumably less stable polymorph, if the latter have obtained their most stable crystal habit [11].

Herein, this phenomenon and experimental evidence establishing this relationship between crystal habit and the ORS are further elucidated. This is achieved using a model system: calcium carbonate (CaCO₃), under anoxic (oxygen depleted) and sweet (carbon dioxide saturated) solution conditions, with cationic additives. CaCO₃ has three polymorphs [12]: calcite, the thermodynamically most stable polymorph with a trigonal crystal structure; aragonite, the second most stable polymorph with an orthorhombic crystal structure [13]; and vaterite, the least stable polymorph with a hexagonal crystal structure [14]. Moreover, with the use of cationic additives, it is possible to achieve a range of crystal habits within individual polymorphs. In this context, the 'prematureearly' nucleation of calcite, in the presence of certain additves, is demonstrated as the trigger for a cascade of transformations, commencing with a star-shaped calcite crystal, moving to aragonite and subsequently to rhombohedral calcite.

Calcium carbonate is widespread, from geological formations to living organisms [15]. It is also of tremendous industrial importance, contributing to scale formation in process equipment [16,17], especially those operating under sweet conditions (i.e. in the presence of carbon dioxide saturated flows). Industrially, it is not uncommon for the nucleation and growth of calcium carbonate to take place in the presence of different cationic species. Understanding the mechanisms by which these cations influence the crystallization of CaCO₃ is crucial for the development of scale mitigation strategies.

The role of cationic additives in the crystallization of $CaCO_3$ has been a subject of intense investigation [18,–20]. The majority of these studies focus on low $S_{CaCO_3}$ conditions, most commonly aqueous solutions containing small concentrations of carbonate and bicarbonate anions. The role of ion adsorption in crystallization of $CaCO_3$ has been probed, where influence in the nucleation step has been noted. In a calcite-forming system, when specific cations are introduced not only does aragonite emerge but also it presents itself as the stable alternative both to calcite and cocrystals (containing $Ca^{2+}$ and the cation additive) [21,22]. Therefore, in the presence of such cations, the formation of aragonite will be favoured. Despite the plethora of investigations on the role of cationic additives in the formation of $CaCO_3$, to our knowledge, there has been a rather small number of studies on the role of $Fe^{2+}$ and $Fe^{3+}$ ions, in particular. One such report highlighted that the cations adsorb on the surfaces of the seeds, thereby inhibiting crystal ($CaCO_3$) growth [22]. Other published literature [23,24] has also reported the addition of $Fe^{2+}$ and $Fe^{3+}$ can serve to be an effective inhibitor of calcite nucleation. This behaviour has been attributed to a decrease in $S_{CaCO_3}$ in the presence of these cations, as compared to a neat solution.

Fewer studies are dedicated to the crystallization of $CaCO_3$ at very high $S_{CaCO_3}$, in sweet and anoxic solution conditions. These conditions, while less popular in academic literature, are highly relevant both industrially and towards understanding natural systems.

Notably, $CO_2$ saturated flows are commonly encountered in the oil and gas industry; deployment of $CO_2$ capture and storage has further increased the prevalence of these conditions. Additionally, naturally occurring relevant systems include existing $CO_2$ rich springs and prehistoric environments (during oceanic formation) where Earth's atmosphere was far richer in $CO_2$. As such these $CaCO_3$ formation conditions, $CO_2$ rich and $O_2$ lean aqueous solutions, have broad reaching implications in understanding of carbonates on early Earth [25].

Herein, an experimental investigation considers the crystallization of calcium carbonate, as a function of $S_{CaCO_3}$, in the absence of any additives, in deaerated, sweet solutions, at high supersaturations. Subsequently, the influence of 1.5 mM of $Fe^{2+}$ in $CaCO_3$ supersaturated solutions is considered. Industrially, this mimics a scenario where corrosion has released $Fe^{2+}$ into the system. Not only is the impact notable, but the findings enable us to draw important conclusions on the role of additives under these conditions. At the same time, a mechanism enabling the isolation of a metastable star-shaped habit of calcite crystals is observed. The study was then expanded to investigate the influence of other cations: $Ba^{2+}$, $Fe^{3+}$, $Li^+$, $Ni^{2+}$ and $Zn^{2+}$.

## 2. Experimental

All the experiments were conducted in round bottom flasks containing 250 ml of deionized water at 80°C, which were de-aerated for 3 h with a $N_2$ stream, at 200 ml min$^{-1}$ (1 bar). Then, $CO_2$ was introduced at 200 ml min$^{-1}$ (0.5 bar) for 12 h. The pH of the solution was adjusted to around 7.2 using 1.05 g of $NaHCO_3$ (Sigma Aldrich). The required $S_{CaCO_3}$ was achieved with the addition of $CaCl_2 \cdot 2H_2O$ (Sigma Aldrich). The pH meters were conducted using the InLab Expert Pro (Mettler Toledo). The concentration of calcium chloride used at different experiments is shown in table 1, along with the concentrations of the different compounds, which have been used in order to investigate the effects of cationic additives. Purging with $CO_2$ was continued throughout the experiment at 200 ml min$^{-1}$; thus the solution remains oxygen free. At the conclusion of each experiment, the solution was filtered to separate solids greater than 3 µm. The dried solid products were assessed using X-ray powder diffraction (XRPD). Measurements reported in this paper were all conducted using an X'Pert PRO diffractometer using Cu-K$\alpha$ radiation (Malvern PANalytical) and scanning electron microscope (SEM) images were taken using a Hitachi TM-100 table-top SEM (Hitachi Ltd). The XRPD measurements were conducted at a $2\theta$ range from 5° to 80°, with a step size of 0.01° ($2\theta$) and a count time of one second. A back-loaded sample holder was used. As the material was relatively free flowing and fine, no mechanical force was exerted on it, to assist in the packing. Thanks to the fine nature of the particles, the surface of the sample can be considered flat with respect to the sample holder. For the crystals produced at 5 min, in the presence of $Fe^{2+}$ and $Ni^{2+}$, the material was unloaded from the sample holder,

**Table 1.** Summary of the experiments conducted in this paper.

| Experiment | CaCl$_2$ (mg ml$^{-1}$) | 1.5 mM additive |
|---|---|---|
| A1 | 2.00 ($S_{CaCO_3}$ ca 2500) [13,14] | — |
| A2 | 8.00 ($S_{CaCO_3}$ ca 10 000) [13,14] | — |
| A3 | 24.00 ($S_{CaCO_3}$ ca 30 000) [13,14] | — |
| B1 | 8.00 | 0.20 mg ml$^{-1}$ of FeCl$_2$ |
| B2 | 2.00 | 0.20 mg ml$^{-1}$ of FeCl$_2$ |
| B3 | 3.00 | 0.20 mg ml$^{-1}$ of FeCl$_2$ |
| B4 | 4.00 | 0.20 mg ml$^{-1}$ of FeCl$_2$ |
| B5 | 16.00 | 0.20 mg ml$^{-1}$ of FeCl$_2$ |
| B6 | 24.00 | 0.20 mg ml$^{-1}$ of FeCl$_2$ |
| B7 | 44.00 | 0.20 mg ml$^{-1}$ of FeCl$_2$ |
| C1 | 8.00 | 0.34 mg ml$^{-1}$ of BaCl$_2$ |
| C2 | 8.00 | 0.26 mg ml$^{-1}$ of FeCl$_3$ |
| C3 | 8.00 | 0.07 mg ml$^{-1}$ of LiCl |
| C4 | 8.00 | 0.21 mg ml$^{-1}$ of NiCl$_2$ |
| C5 | 8.00 | 0.22 mg ml$^{-1}$ of ZnCl$_2$ |

mixed with the mother batch and then a new sample was taken for a second measurement. The same procedure was repeated once again. The three measurements were identical.

# 3. Results

## (a) Crystallization of calcium carbonate at high $S_{CaCO_3}$

Three baseline experiments were conducted (table 1, A1–A3), using CaCl$_2$·2H$_2$O: 2 ($S_{CaCO_3}$ ca 2500), 8 ($S_{CaCO_3}$ ca 10 000) and 24 mg ml$^{-1}$ ($S_{CaCO_3}$ ca 30 000) [14,15]. In all three cases, vaterite emerges first; it then gradually dissolves coincidentally with aragonite formation (figure 1). The presence of calcite is not observed under all conditions tested. At the lowest supersaturation considered, $S_{CaCO_3}$ ca 2500, vaterite crystals with a snowflake shape are apparent at 15 min (figure 2$a$). The dissolution of aragonite needles is still observed at 96 h (figure 2$c$).

By contrast, upon increasing the $S_{CaCO_3}$ to ca 10 000 (corresponding to a calcium chloride concentration of 8 mg ml$^{-1}$), needle-shaped aragonite crystals begin to appear in 1 h, alongside flower-shaped vaterite crystals (figure 2$d$); simultaneously, a rhombohedral calcite crystal is also observed, along with flower-shaped vaterite crystals as well (figure 2$e$). The appearance of calcite signals that the system has reached the solubility limit for CaCO$_3$. A further increase in $S_{CaCO_3}$, to ca 30 000, demonstrates a completed polymorphic transformation to the stable calcite polymorph within 96 h (figure 2$f$), reaching the end of the ORS.

## (b) Crystallization of calcium carbonate in the presence of 1.5 mM of Fe$^{2+}$

Comparable experiments were conducted with the addition of 1.5 mM Fe$^{2+}$ (table 1, B1–B7). For 8 mg l$^{-1}$ with 1.5 mM of Fe$^{2+}$ ($S_{CaCO_3}$ ca 10 000 with no Fe$^{2+}$) only calcite is identified at 15 min (figure 3), whereas the predominant polymorph at the same time without the addition of Fe$^{2+}$ was vaterite. At this early stage, the calcite polymorph has a distinct star-shaped habit (figure 4$a$). The formation, i.e. 'premature', star-shaped crystals has been observed (figure 4$b$). During the growth

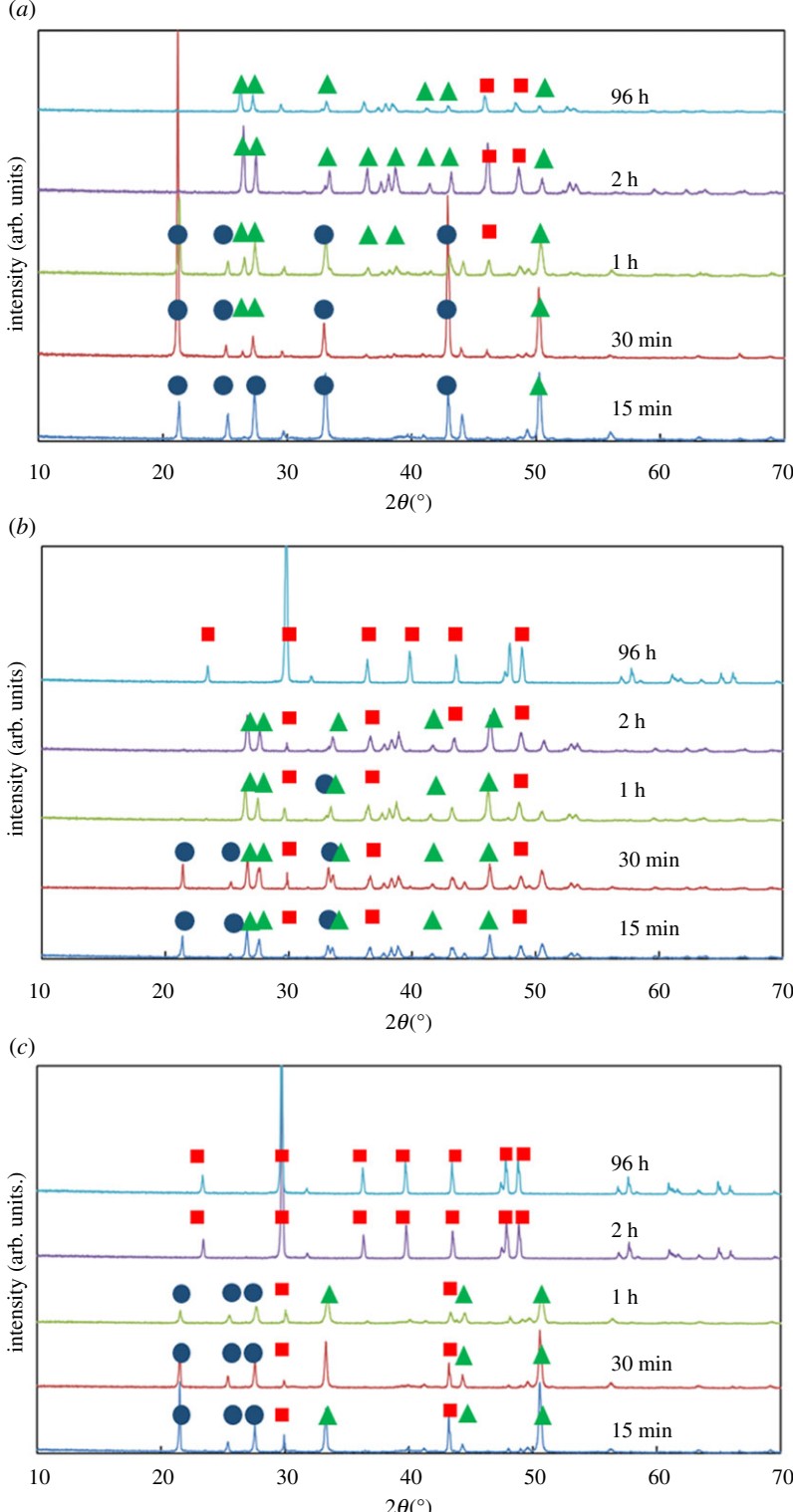

**Figure 1.** The XRPD patterns for the powder obtained from the crystallization of calcium carbonate at different durations (shown in the legend) at different concentrations of calcium chloride: (*a*) 2 mg ml$^{-1}$, (*b*) 8 mg ml$^{-1}$, (*c*) 24 mg ml$^{-1}$. Blue circles indicate the characteristic peaks of vaterite, green triangles the characteristic peaks of aragonite and red squares the characteristic peaks of calcite. (Online version in colour.)

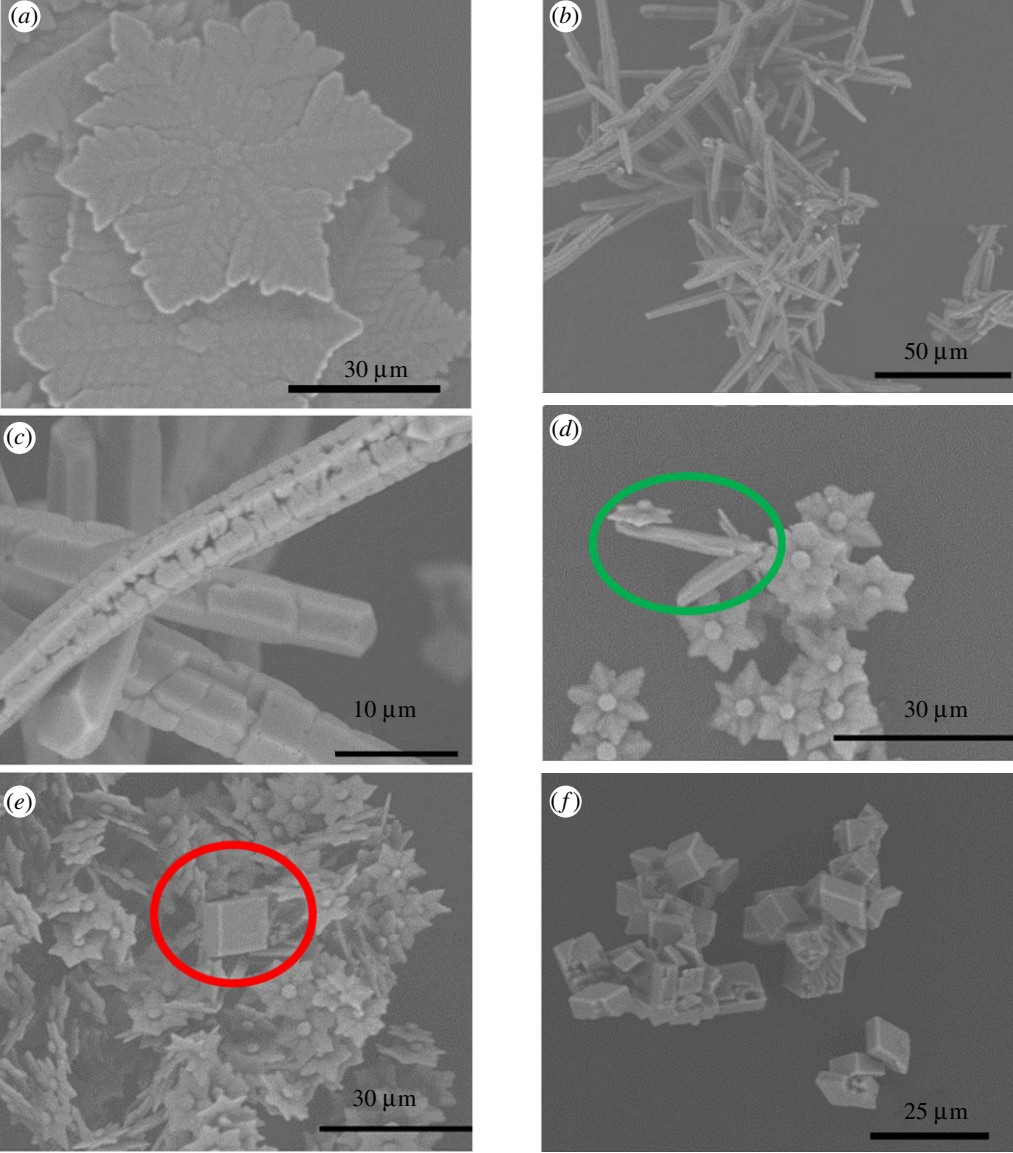

**Figure 2.** SEM images obtained from crystallization experiments conducted at: (*a*) a calcium chloride concentration of 2 mg ml$^{-1}$ for 15 min, (*b,c*) a calcium chloride concentration of 2 mg ml$^{-1}$ for 96 h, (*d,e*) a calcium chloride concentration of 8 mg ml$^{-1}$ after 1 h (the green circle indicates needle-shaped aragonite crystals and the red circle indicates a rhombohedral calcite crystal) and (*f*) a calcium chloride concentration of 8 mg ml$^{-1}$ after 96 h. (Online version in colour.)

of these star-shaped crystals, elucidated from SEM images and XRPD patterns (figure 3) for times less than 15 min, no evidence of any other crystal habits or polymorphs has been identified.

With increasing time, aragonite, with a needle-like crystal habit, emerges as the dominant polymorph. Qualitatively this is observed in SEM images, as well as semi-quantitatively in the XRPD patterns (figure 3). During the relevant time-window (30 min–2 h), an increasingly strong aragonite signal is observed (e.g. 2$\theta$ of 26.25° and 27.25°). During this period (notably at 1 h), the simultaneous apparent dissolution of the star-shaped calcite crystals commences (figure 4$c$).

To accelerate the progression of the ORS, an increased concentration of Ca$^{2+}$, with 1.5 mM Fe$^{2+}$, was studied. As can be seen in figure 4$d$, after 48 h, both star-shaped calcite crystals and

**Figure 3.** XRPD patterns obtained from crystallization experiments at a calcium chloride concentration of 8 mg ml$^{-1}$, in the presence of 1.5 mM of Fe$^{2+}$, at different times. The green triangles indicate the characteristic peaks of aragonite, the open red squares demarcate the characteristic peaks of calcite for the cases where the SEM images suggest the presence of star-shaped calcite crystals and the half-filled red squares represent the peaks of calcite for the cases where SEM images suggest the coexistence of star-shaped and rhombohedral calcite crystals. (Online version in colour.)

rhombohedral calcite crystals coexist; the star-shaped calcite crystals are dissolving and the rhombohedral calcite crystals are growing. In some instances, rhombohedral crystals (figure 4$d,e$) appear to retain curved edges, inconsistent with the traditional formation of this calcite habit, suggesting that the transition from star-shaped to rhombohedral crystals does not necessarily require the complete dissolution of star-shaped crystals. Elsewhere (figure 4$f$), the calcite crystals seem to have the well-known rhombohedral shape. This thereby suggests that the star-shaped crystals dissolve either completely or partially. For the latter, the dissolution would yield multifaceted entities, which regrow into rhombohedral-shaped crystals. It must also be noted that no FeCO$_3$, commonly known as siderite, was observed by either XRPD or SEM.

## (c) Crystallization of calcium carbonate in the presence of other cations

Experiments were performed, as described above, with 1.5 mM of other cation additives (table 1, C1–C5): Ba$^{2+}$, Fe$^{3+}$, Li$^+$, Ni$^{2+}$, Zn$^{2+}$. The changes to CaCO$_3$ supersaturation are assumed to be small, with the exception of Ba$^{2+}$ [26]. In the presence of Ba$^{2+}$, crystallization does not follow the ORS. Rhombohedral calcite emerges as the only polymorph after the very rapid onset of crystallization in less than 15 min (figure 5$a$).

With the addition of either Fe$^{3+}$ or Li$^+$, the polymorphism evolves as was observed in the absence of any other cations: vaterite to aragonite, and then to rhombohedral calcite (figure 5$b,c$).

Neither BaCO$_3$ nor LiCO$_3$ (nor any other carbonate-containing products) are detected in the experiments when either Ba$^{2+}$ or Li$^+$ is present, respectively. However, a red precipitate, attributed to amorphous Fe(OH)$_3$, was observed when FeCl$_3$ was used. This side reaction reduces the concentration of Fe$^{3+}$ in solution. Therefore, the change in chemical potential of the solution, caused by the addition of Fe$^{3+}$, is small, and the S$_{CaCO_3}$ remains high.

The formation of an additional product is notable because it likely influences a more rapid progression through the ORS. As such, the polymorphic transformations are, therefore, faster in the presence of Fe$^{3+}$ rather than in the presence of Li$^+$. The progression of crystalline species is traced through XRPD patterns at different times (figure 5$b,c$). In contrast to the case with Li$^+$

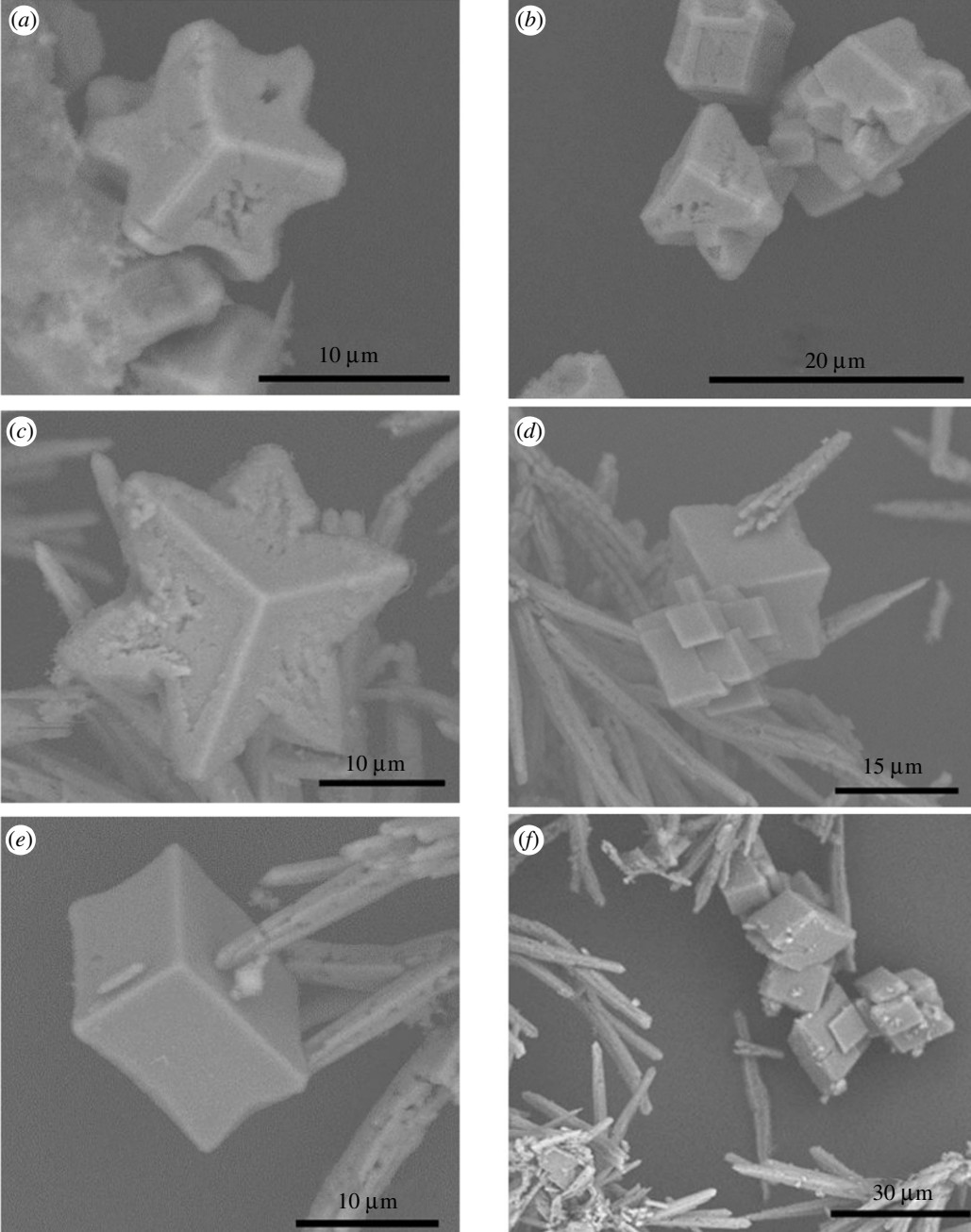

**Figure 4.** SEM images of crystals obtained from crystallization experiments at (*a,b*) calcium chloride concentration of 8 mg ml$^{-1}$, in the presence of 1.5 mM of Fe$^{2+}$, after 15 min, (*c*) calcium chloride concentration of 8 mg ml$^{-1}$, in the presence of 1.5 mM of Fe$^{2+}$, after 1 h, and (*d–f*) calcium chloride concentration of 24 mg ml$^{-1}$, in the presence of 1.5 mM of Fe$^{2+}$, after 48 h. (Online version in colour.)

which has vaterite crystals after 2 h, in the presence of Fe$^{3+}$, after 2 h, the transformation to calcite is complete.

It should be noted, however, that the polymorphic transformations in the presence of either Fe$^{3+}$ or Li$^+$ are significantly slower than the comparable cases in the absence of the cations (at the same concentrations of Ca$^{2+}$).

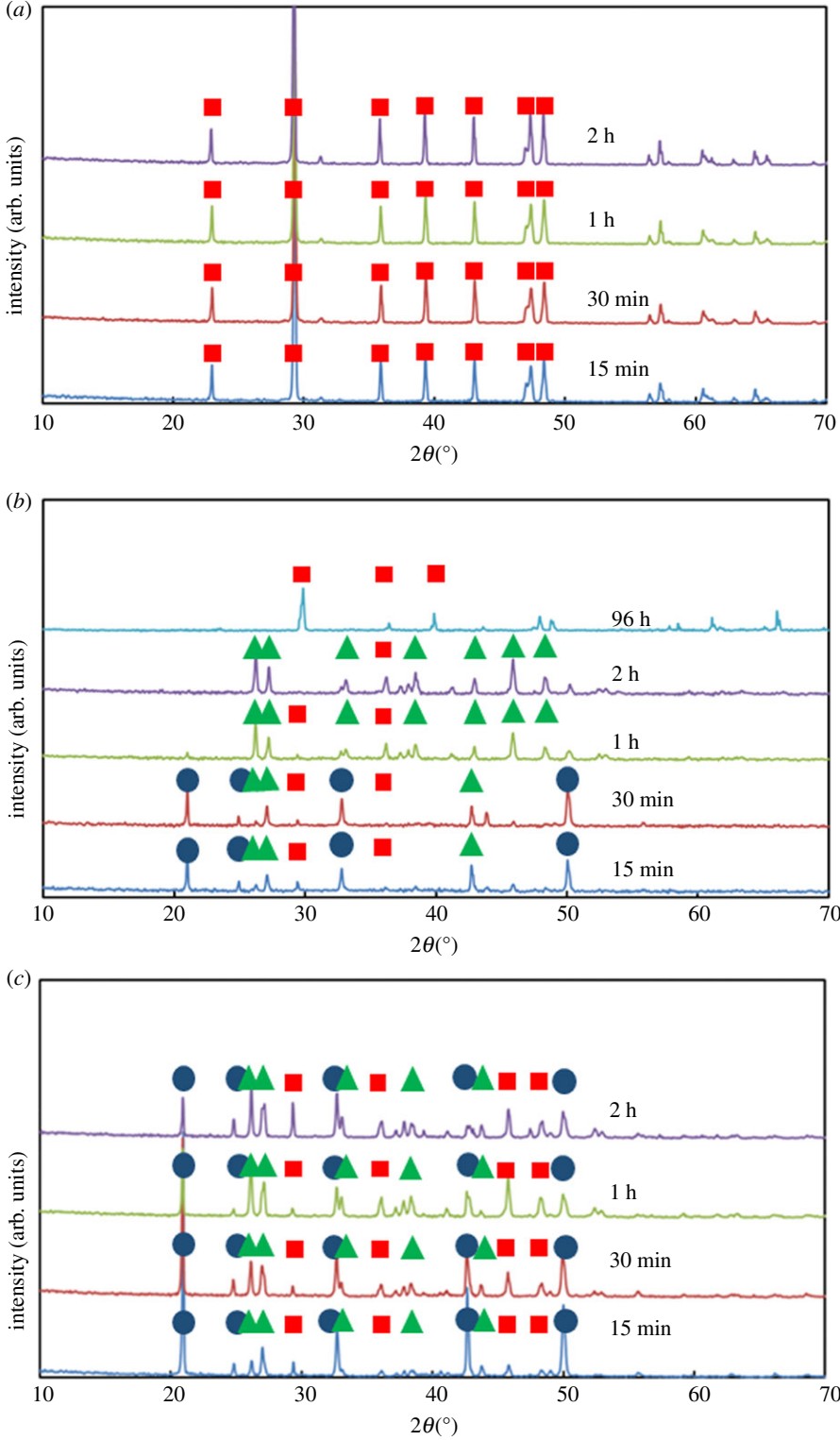

**Figure 5.** The XRPD patterns for the powder obtained from the crystallization of calcium carbonate at different durations (shown in the legend) at 8 mg ml$^{-1}$ of calcium chloride in the presence of 1.5 mM of: (a) $Ba^{2+}$, (b) $Fe^{3+}$, (c) $Li^{+}$. Blue circles indicate the characteristic peaks of vaterite, green triangles the characteristic peaks of aragonite and red squares the characteristic peaks of calcite. (Online version in colour.)

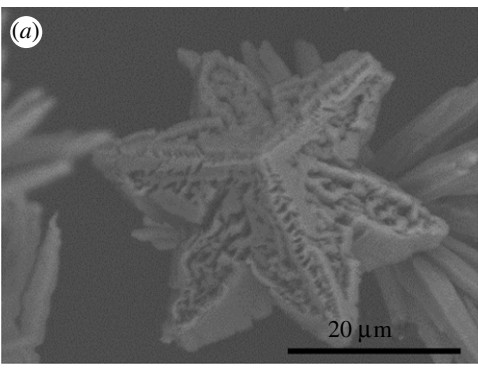
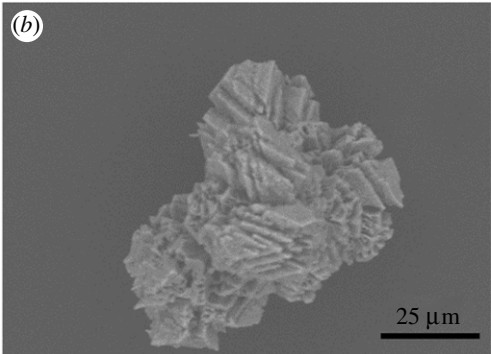

**Figure 6.** SEM images of crystals obtained from crystallization experiments with calcium chloride concentration of 8 mg ml$^{-1}$, in the presence of 1.5 mM of Ni$^{2+}$, after (*a*) 15 min and (*b*) 30 min.

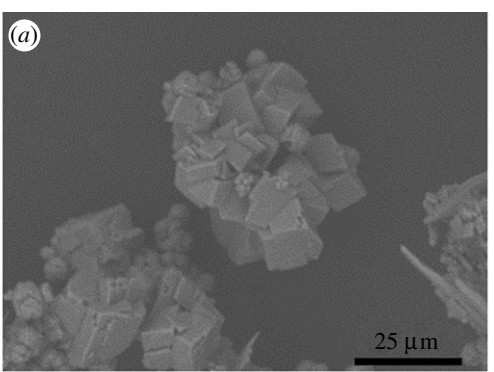
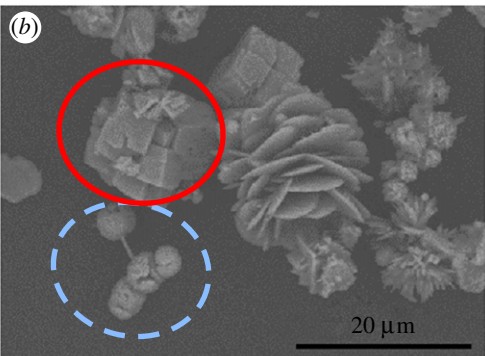

**Figure 7.** SEM images of crystals obtained from crystallization experiments with calcium chloride concentration of 8 mg ml$^{-1}$, in the presence of 1.5 mM of Zn$^{2+}$, after (*a*) 15 min and (*b*) 1 h. The red circle indicates rhombohedral calcite crystals, while the dotted brown circle indicates spherulites of smithsonite (ZnCO$_3$). (Online version in colour.)

The addition of Ni$^{2+}$ is similar to that of Fe$^{2+}$. That is, star-shaped calcite crystals (figure 6*a*) emerge first, followed by aragonite crystals. The metastability of these stars can be clearly evidenced in imaging showing them in the process of dissolution (figure 6*b*). The dissolution pattern of the star-shaped calcite crystals clearly demonstrates that the evolution is not directly to rhombohedral calcite.

In the presence of Zn$^{2+}$, a breadth of species is observed. Early on, poorly specified (i.e. non-rhombohedral) calcite emerges as an aggregate of particles (figure 7*a*). Alongside calcite, cauliflower-shaped aragonite, vaterite and spherulitic particles of smithsonite (ZnCO$_3$) (in the dotted circle in figure 7*b*) [27] are also observed. With longer crystallization times, rhombohedral calcite crystals are evident (figure 7*b*). Other changes in the crystal structures are also observed: vaterite crystals start as platelets and later assemble into flower-shaped aggregates (figure 7*b*). This crystal habit has been previously identified in the presence of highly charged amphiphilic molecules or CH$_3$NH$_2{}^+$ ions [20]. Smithsonite also appeared on the corresponding XRPD pattern and spherulitic particles have been previously reported [28].

## 4. Discussion

Nucleation kinetics have been used to explain the high supersaturation prerequisite required to trigger the ORS [29,30]. The magnitude of supersaturation required (to trigger the ORS) strongly depends on the nature of the compound.

Herein, very high supersaturations ($S_{CaCO_3} \geq 2500$) were, initially, explored through the exemplar case of calcium carbonate crystallization in the absence of any additives. Under these conditions, crystallization was observed to evolve through a series of polymorphic transformations. The most metastable polymorph (i.e. least stable), vaterite, emerged first as snowflake-shaped crystals. This unusual crystal habit has only been previously obtained through double diffusion experiments [31]. As the crystallization proceeds, the solution $S_{CaCO_3}$ decreases.

As $S_{CaCO_3}$ decreases, eventually $[Ca^{2+}] \cdot [CO_3^{2-}] < K_{sp'Aragonite}$ (where $K_{sp'Aragonite}$ stands for the solubility product of aragonite) resulting in the dissolution of vaterite crystals, as described by the ORS. Subsequently, needle-shaped aragonite crystals emerge. In the same motif, as the crystallization continues, the $S_{CaCO_3}$ decreases further. Eventually, $[Ca^{2+}] \cdot [CO_3^{2-}] < K_{sp'Calcite}$ (where $K_{sp'Calcite}$ stands for the solubility product of calcite). This leads, in turn, to the dissolution of aragonite crystals and the formation of rhombohedral calcite crystals—the most thermodynamically stable polymorph.

The rate of the ORS events can be controlled by modifying the initial $S_{CaCO_3}$. An increase in $S_{CaCO_3}$ increases the nucleation rate. This increase of the $S_{CaCO_3}$ increases the rate of the polymorphic transitions, which in turn allows for the faster appearance of the stable rhombohedral calcite crystals.

The implications of increasing supersaturation, i.e. speeding-up the appearance of stable rhombohedral calcite crystals, can be considered in the context of the relative magnitude of the $S_{CaCO_3}$. Firstly are the cases with very low $S_{CaCO_3}$ (which can be driven both by the low concentration of $Ca^{2+}$ and $CO_3^{2-}$, but also from the presence of cationic additives). There, calcite is the sole polymorph that will emerge as the conditions for ORS will not be met and therefore the phenomenon will not be triggered. Secondly, there are the cases with intermediate magnitude of $S_{CaCO_3}$, where the conditions of ORS are met. Under these conditions, ORS does occur, but at a very slow rate. Therefore, stable rhombohedral calcite crystals will not be observable during a reasonable time frame, such as less than 48 h; although they should be observed with increased timescales. During an extended period of time, such as more than 48 h, aragonite crystals will be, initially, predominant with increasing presence of rhombohedral calcite crystals. Thirdly, for cases with elevated $S_{CaCO_3}$, the ORS phenomenon does occur very quickly. As the rate of transformations will be really fast, stable rhombohedral calcite crystals emerge relatively quickly. These scenario are all supported by the previously presented experimental results in figure 1. In short, the highest and lowest $S_{CaCO_3}$ give stable rhombohedral calcite crystals, whereas the intermediate $S_{CaCO_3}$ produces primarily aragonite crystals.

The three cases, of different relative values of $S_{CaCO_3}$, were further explored using a range of $Ca^{2+}$ concentrations with the addition of 1.5 mM of $Fe^{2+}$ (table 1, B1–B7). Experiments were conducted where $Ni^{2+}$ was used instead of $Fe^{2+}$ with very similar outcomes. For both the lowest (2 and 3 mg ml$^{-1}$) and highest (44 mg ml$^{-1}$) concentrations of $CaCl_2$, rhombohedral calcite crystals predominated after 48 h (figure 8). By contrast, crystals obtained from solutions with intermediate concentrations (4–44 mg ml$^{-1}$) were a mixture comprising needle-shaped aragonite crystals, star-shaped calcite crystals and rhombohedral calcite crystals (figure 8). Under the investigated conditions, the presence of $Fe^{2+}$ does not seem to promote the formation of aragonite via inhibition of the nucleation and/or growth of calcite crystals.

The observations strongly support the supersaturation-dependent triggering of the ORS. At the very low concentrations (2 and 3 mg ml$^{-1}$), the ORS was not triggered and only rhombohedral calcite crystals are observed (figure 9*a,b*) with no relics of any other polymorph. At high supersaturations (44 mg ml$^{-1}$), the polymorphic transformations have proceeded quickly towards stable rhombohedral calcite crystals (figure 9*e,f*). At intermediate supersaturations, both aragonite needles and star-shaped calcite crystals are observed at 48 h (figure 9*c,d*). To confirm the explanation for the failure to trigger ORS at low concentrations, additional experiments (calcium chloride $\leq$ 3 mg ml$^{-1}$) were carried at 15, 30, 60 and 120 min. The results confirmed that no star-shaped calcite crystals emerge. Similarly, crystallization at high concentrations (calcium chloride at 44 mg ml$^{-1}$), i.e. undergoing rapid polymorphic transformation, was similarly probed at short

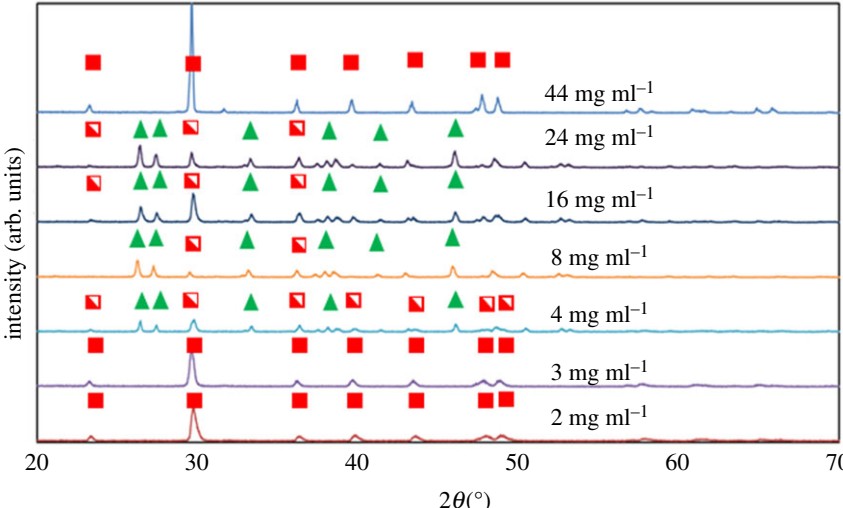

**Figure 8.** XRPD patterns obtained from crystallization experiments conducted using the concentrations of calcium chloride shown in the legend, in the presence of 1.5 mM of $Fe^{2+}$, for 48 h. Green triangles stand for the characteristic peaks of aragonite. Filled red squares stand for the characteristic peaks of calcite, in cases where SEM images suggest the existence of rhombohedral calcite crystals only. The half-filled red squares represent the peaks of calcite for the cases where SEM images suggest the coexistence of star-shaped crystals with rhombohedral crystals. (Online version in colour.)

times (15 and 30 min). The emergence of star-shaped crystals (which eventually dissolve at the expense of rhombohedral calcite crystals) was confirmed. Another interesting observation has to do with the fact that $Ba^{2+}$ seems to prevent completely the occurrence of ORS.

In a recent paper, Barlow *et al.* elaborated, starting from a mathematical description of the transformations occurring during ORS, on the conditions required for the ORS to be observable [32]. Their findings suggest that for the ORS cascade of transformations to be observable, the rate constant of the first phase transition needs to be the dominant/fastest one. Applying the notions developed in that work in the current paper, some important conclusions can be deduced. At low $S_{CaCO_3}$, in the absence of any additives, the kinetics of vaterite and aragonite formation are way too slow. Similarly, the presence of $Ba^{2+}$ seems to reduce $S_{CaCO_3}$ so much that again the kinetics for vaterite and aragonite are diminished.

The observation of star-shaped calcite crystals emerging first, at particular values of $S_{CaCO_3}$ in the presence of $Fe^{2+}$ ions, and their subsequent dissolution, in favour of aragonite crystals, is a key outcome of this work. The dissolution of the star-shaped calcite crystals, at the expense of aragonite, is indicative that this crystal habit, of calcite, must have a higher apparent solubility than needle-shaped aragonite. This unexpected outcome is the result of a higher apparent solubility for the star-shaped calcite crystals, which is attributed to the higher surface energy of their crystal facets. The authors have already demonstrated this difference in surface energy, in a recent paper [33]. As such, the distinction between the two habits of calcite suggest that the star-shaped calcite crystals should be classified as metastable. This classification is with respect to not only the needle-shaped aragonite crystals, but also the rhombohedral calcite crystals. This observation complicates the supposition that under ORS, crystals are driven towards the thermodynamically stable polymorph. Previously, no regard for the crystal habit has been included. The evidence herein suggests that the ORS drives the crystyallization towards not only the most stable polymorph but also towards the most stable crystal habit.

The $Fe^{2+}$ ions, in the presence of $CO_2$-derived species, have the potential of forming ferrous carbonate, siderite. Notably, siderite exhibits the same crystal structure as calcite (and gaspeite, in the case of $Ni^{2+}$, which also has a trigonal crystal structure like calcite). Previous crystallization

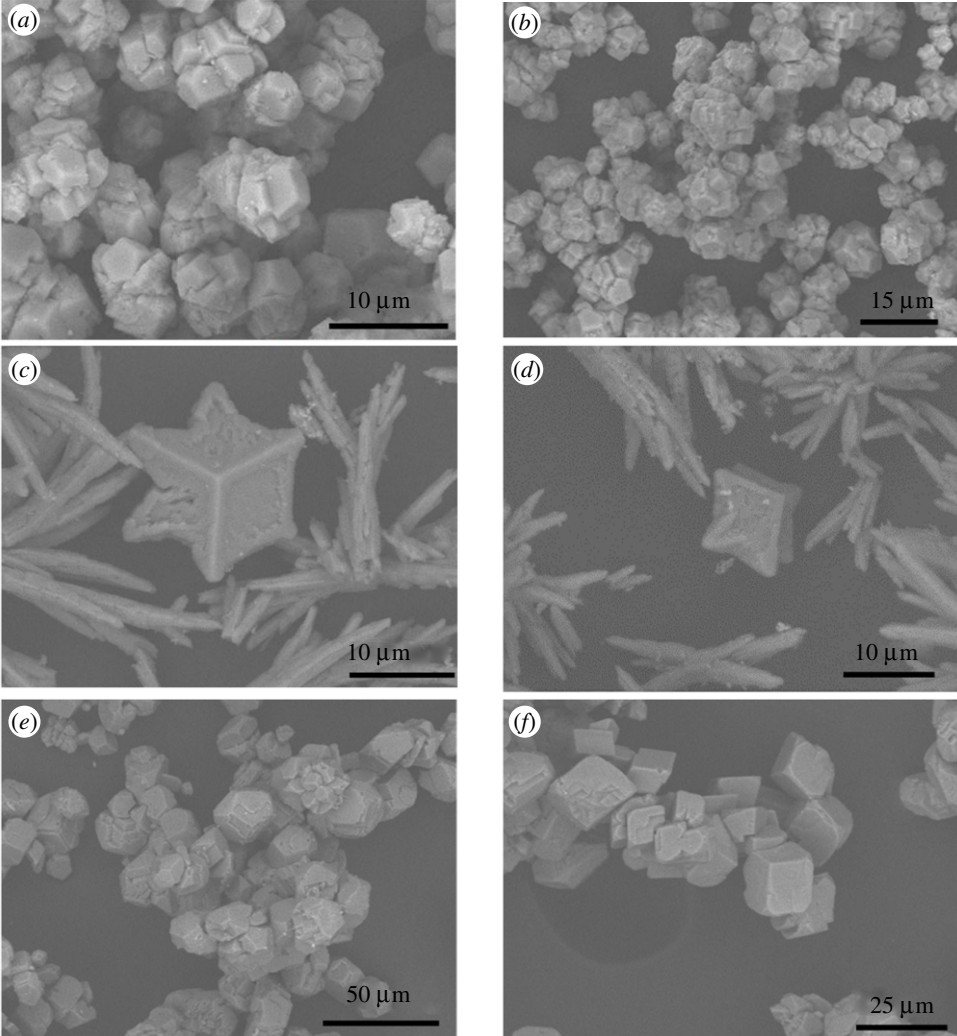

**Figure 9.** SEM images of crystals obtained from crystallization experiments with: (*a,b*) 2 mg ml$^{-1}$ of calcium chloride, in the presence of 1.5 mM of Fe$^{2+}$, after 48 h, (*c,d*) 16 mg ml$^{-1}$ of calcium chloride, in the presence of 1.5 mM of Fe$^{2+}$, after 48 h and (*e,f*) 44 mg ml$^{-1}$ of calcium chloride, in the presence of 1.5 mM of Fe$^{2+}$, after 48 h.

studies, at much lower supersaturations, have suggested that the presence of such cations favours the formation of aragonite. The experiments conducted herein suggest that this behaviour is condition-specific, as these same results were not observed in this work. Rather, herein star-shaped calcite crystals emerge before aragonite. For the calcite crystals obtained in the presence of 1.5 mM of Fe$^{2+}$ after 15 min of crystallization, no traces of aragonite have been detected by means of XRPD, SEM or Fourier transform infrared (FTIR) spectroscopy [11].

In previous studies, it has been suggested that the presence of Fe$^{2+}$ ions (as well as of Fe$^{3+}$), at low supersaturations, leads to the inhibition of the growth of calcite seeds. In those low supersatutations [22,23,34], the inhibition required only a few micromoles of Fe$^{2+}$; the inhibitive properties are observed to be further enhanced in the presence of dissolved oxygen. Those works suggest that the presence of Fe$^{3+}$ ions severely inhibits the growth of calcite, by means of adsorption on specific crystal facets. The findings of this paper show that this effect decreases with increasing S$_{CaCO_3}$ (provided that the concentration of Fe$^{3+}$ remains constant).

Evidence presented in this work (including XRPD patterns from figure 1b and figure 5b, as well as SEM images) only identifies small differences in the crystals (polymorphism, crystal habit and size) produced using $8 \, \text{mg ml}^{-1}$ of $CaCl_2$, in the presence and the absence of 1.5 mM of $Fe^{3+}$ ions. This occurs because $Fe^{3+}$ ions precipitate in solution, creating an amorphous hydroxide ($Fe(OH)_3$). The $Fe^{3+}$ ions need to be present in the solution in the form of free ions, in order to contribute to the kinetics of the ORS. The fact that a large number of $Fe^{3+}$ ions are consumed in the formation of the hydroxide means that $Fe^{3+}$ ions are not available to contribute to the ORS. By contrast, $Li^+$ ions remain in solution; no precipitation of a lithium salt is observed. Therefore, they modify the chemical potential of the solution, changing the kinetics of the ORS. It was demonstrated that cations, only in their free form, decrease the $S_{CaCO_3}$ and slow the kinetics of the ORS. This behaviour is consistent with existing theory based on similar phenomena involving different cationic species. It should be noted that various factors, including the size and the charge of the cations, determine influence the chemical potential differently; an *a priori* prediction of chemical potential, for different arbitrary systems, is not yet possible. Therefore, extrapolation of the phenomena reported in the previous study to the conditions herein appears to be inapplicable.

Star-shaped calcite crystals emerge in the presence of $Fe^{2+}$ and $Ni^{2+}$ cations, both of which have the ability to form carbonates with the same crystal structure as calcite [33]. In both cases, the star-shaped crystals emerge before aragonite.

It can be argued that $Fe^{2+}$ and $Ni^{2+}$ inhibit the nucleation and/or growth of aragonite, early on. Thus, calcite emerges as the dominant polymorph and the star-shaped morphology can then be attributed to preferential adsorption of $Fe^{2+}$ ions on specific facets. However, this explanation is not consistent with other reports from the literature which suggest that the presence of $Fe^{2+}$ and $Ni^{2+}$ ions suppresses the formation of calcite altogether, favouring aragonite formation instead. Furthermore, it fails to explain why the star-shaped calcite crystals dissolve and later rhombohedral calcite crystals grow, instead of growing to rhombohedral calcite crystals at the outset.

An alternative explanation is that the star-shaped crystals are, in fact, cocrystals/solid solutions of $Ca^{2+}$ with either $Fe^{2+}$ or $Ni^{2+}$. The star-shaped cocrystals are metastable with respect to aragonite and as the crystallization occurs through the ORS they emerge first. This explanation is not fully satisfactory, as no experimental evidence (neither XRPD patterns nor FTIR spectra) have suggested the formation of such cocrystals. Furthermore, vapour pressure measurements [33] did not provide definitive evidence that the higher apparent solubility of the star-shaped calcite crystals is because of the formation of solid solutions/cocrystals. Besides, an added challenge is that no star-shaped calcite crystals or any other high apparent solubility calcite crystal habits are observed for other case studies (such as with $Zn^{2+}$ or $Mg^{2+}$) where the literature has previously indicated a penchant for the formation of cocrystals in relevant $Ca^{2+}$ solutions.

A final possible explanation is that the presence of either $Fe^{2+}$ or $Ni^{2+}$ cations may promote the nucleation of calcite, at the expense of aragonite, under some conditions. As most literature has focused on lower supersaturation solutions, the observation of alternative crystal habits is unique to these conditions. Moreover, the formation of the star-shaped calcite crystals occurs where the supersaturation in the bulk solution is very high. And consistent with the literature, the formation of rhombohedral calcite crystals has been observed when the supersaturation in the bulk solution was reduced, to levels more consistent with the literature. As such, a metastable crystal habit can emerge thanks to the high supersaturation.

Putting the findings from $Fe^{2+}$ and $Ni^{2+}$, suggesting the occurrence of star-shaped calcite crystals, in the scrutiny of the work by Barlow *et al.* there seem to be some interesting implications [31]. The presence of these additives could favour the kinetics of the formation of star-shaped calcite or cocrystal. This may be done by means of lowering the energetic barrier for formation of calcite or of the hypothetical cocrystal. Template-assisted nucleation is well known to provide access to certain polymorphs by similar means.

It has been noted, both in the Results section and briefly in the Discussion, that the lifecycle of the crystals observed in the experiments conducted herein shows a pattern of dissolution

and formation [35]. That is, the emergence of a stable crystal habit is from a sequence of dissolution (complete or partial) and formation steps. This is in direct contrast to the alternative behaviour whereby changes in crystal habit, towards the metastable form, result from facet specific growth. This distinction verifies that the surface properties are a crucial descriptive factor in the determination of the identity of individual particles. Surface properties, stemming from the differences in crystal habit, should be considered when the dissolution properties are a matter of concern. Thus, it will be particularly interesting to re-examine, in the light of the findings of this work, the findings of experiments conducted to assess the dissolution performance and the stability of crystalline materials [36].

However, the most noteworthy finding of this work is the role of the 'early' nucleation in the occurrence star-shaped calcite crystals. This phenomenon triggers the cascade of transformations, described in detail in this paper, for the first time. This feature is controlled by the additives and it can be correlated with the ability of certain additives to form carbonates with the same crystal structure as calcite: trigonal.

In areas like the pharmaceutical industry, where particle morphology modifications can be used as a vehicle to improve the solubility of hydrophobic active pharmaceutical ingredients, this finding can allow the design and production of particles with exotic morphologies, debottlenecking development activities. This could be done in synergy with the already established field of nucleation control; especially considering more recent advances in the development of soft templates.

## 5. Conclusion

The crystallization of $CaCO_3$ under sweet and anoxic conditions and elevated $S_{CaCO_3}$ obeys the ORS. The introduction of cations which can form carbonates with a trigonal crystal structure, such as $Fe^{2+}$ and $Ni^{2+}$, may facilitate the nucleation of calcite, the trigonal polymorph of calcium carbonate, prior to the emergence of aragonite.

Notably, the 'early' nucleation of calcite and the subsequent growth under high supersaturation lead to the formation of calcite crystals with high surface energy, star-shaped, crystal habits, which are newly observed in this work. This capability, to use additives for the generation of stable polymorphs with metastable crystal habits, could be harnessed for particle engineering operations.

Data accessibility.  Raw data for the XRD patterns can be found in the electronic supplementary material [37].

Authors' contributions.  E.H.: conceptualization, data curation, formal analysis, investigation, methodology, validation, writing—original draft, writing—review and editing; S.M.V.: conceptualization, funding acquisition, methodology, project administration, supervision, writing—review and editing; J.D.L.: validation, writing—review and editing; K.L.S.C.: funding acquisition, investigation, methodology, supervision, writing—original draft, writing—review and editing.

All authors gave final approval for publication and agreed to be held accountable for the work performed therein.

Competing interests.  We declare we have no competing interests.

Funding.  The authors would like to acknowledge the funding and technical support from BP through the BP International Center for Advanced Materials (BP-ICAM), which made this research possible. This work was also partly funded by the EPSRC through a Prosperity Partnership grant (EP/R00496X/1).

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
