## [Peer Review File · Proceedings. Mathematical, Physical, and Engineering Sciences]

Review History

RSPA-2021-0601.R0 (Original submission)

Review form: Referee 1

Is the manuscript an original and important contribution to its field?

Acceptable

Is the paper of sufficient general interest?

Acceptable

Is the overall quality of the paper suitable?

Acceptable

Can the paper be shortened without overall detriment to the main message?

Yes

Do you think some of the material would be more appropriate as an electronic appendix?

No

Do you have any ethical concerns with this paper?

Yes

Recommendation?

Major revision is needed (please make suggestions in comments)

Comments to the Author(s)

This manuscript describes the process of crystallization of CaCO_3 at high supersaturation conditions in the absence as well as in the presence of different inorganic additives, such as Li^+ , Fe^{2+} , Ba^{2+} , Ni^{2+} , Fe^{3+} cations. Within this research Authors have explored the subsequent formation of vaterite, aragonite, and calcite, exhibiting different crystal habits and studied the relation between the polymorph stability and its morphology.

Among the most important results, Authors are highlighting two observations. The first one is related to the control over the nucleation rate of CaCO_3 from its pure (not contaminated) solution. Authors show how the changes in the concentration influence the formation and stability of all three above-mentioned polymorphic forms. At low concentrations, they observed sequence of transformation, from the least stable (vaterite) with the snowflake grains, through needle-shaped aragonite, to the most stable rhombohedral calcite, whereas at increased concentration the calcite has been formed with a much faster rate. The second observation is connected to formation of the star-shaped calcite crystals in presence of cations, which can form trigonal carbonates, such as Fe^{2+} and Ni^{2+} . This is presented as a complication to the Oswald rules of Stages, as the process of crystallization is directed not only to the most stable polymorph but also toward the most stable crystal habit of certain polymorphic form. Unfortunately, these findings are overlapping with previously reported by these same Authors in the article published only few months ago (Campbell et al. 2021 Proceedings A) which strongly influences the novelty of current research.

Despite the fact, that Authors has provided sufficient techniques and adequate data to support their observations and the research done within this work is of good and general interest, in my opinion, the manuscript in its current form doesn't meet novelty requirements for Proceedings A. Especially, that the connection between the surface energy of the crystals grains and formation of the star-shaped crystals in the presence of Ni^{2+} was reported in their previous article. I am strongly convinced that this problem can be resolved, by addition of a clear and exhaustive fragment where the novelty of the current report would be expressed in the context of the previous work.

Minor issues:

in Experimental Section:

1. What was the pressure of CO_2 gas used in the experiment?
2. The source of X-ray radiation should be given.
3. How was the pH of the solution been estimated?

Results:

1. In Figures 1a and b there are some unassigned reflections (at ca. 30° , 42°), can Authors indicate their possible origin?
2. Please do replace all 'diffraction peaks' with reflections.
3. Authors are using the results of their FTIR experiments to support their findings (for example line 409 p 27, l. 451 p. 29) but no spectra or information about the instrument are provided. Unfortunately, that seems like some copy/paste leftover from Authors' previous work, where this technique was employed.

Review form: Referee 2

Is the manuscript an original and important contribution to its field?

Good

Is the paper of sufficient general interest?

Excellent

Is the overall quality of the paper suitable?

Excellent

Can the paper be shortened without overall detriment to the main message?

Yes

Do you think some of the material would be more appropriate as an electronic appendix?

No

Do you have any ethical concerns with this paper?

No

Recommendation?

Accept with minor revision (please list in comments)

Comments to the Author(s)

In this manuscript the authors report the result of an interesting collection of experiments on supersaturated CaCO_3 solutions. They clearly show the presence of Ostwald's rule of stages in the growth process and how the presence of various metal cations affect the outcome of the growth process from solution. This paper represent a significant addition to the knowledge base in this field and should be published given the minor corrections listed below.

In the abstract, line 28. Define the notation used to indicate a supersaturated condition in the intro. It is better here to state specifically: " for high relative initial supersaturations such as 2500,...". Additionally, since there are several ways of defining "supersaturation" perhaps this should be defined in the intro section as well.

Also on line 28: Define the terms "sweet" and "anoxic" later in the intro. Here be specific: "Oxygen rich solutions and oxygen depleted conditions..." Here I am speculating this is what is meant by the authors as sweet and anoxic.

Line 64: One might here, at the end of the first sentence, give the definition for initial supersaturation used in this paper and introduce the shorthand notation used by the authors: SCaCO_3

Line 91: This is a good place to define the terms "sweet" and "anoxic".

Line 101: Here the authors belatedly define the term "sweet" as used in this report. Define it earlier in the intro and remove the phrase in parenthesis here.

Line 108: The authors define the shorthand notation for supersaturated CaCO_3 in a brief fashion here. Define in a formal, and complete way earlier in the intro.

Line 112: The sentence that begins on this line ,where the authors make claims about the behavior of “calcite forming systems” upon addition of certain cations could use a reference.

Line 116: The sentence beginning with “Limited additional..” should be revised. Perhaps use: “To our knowledge, there has been a rather small number of studies on the role of additives...”

Further, this is a confusing statement since the authors state on line 106 that “The role of cationic additives in the crystallisation of CaCO₃ has been a subject of intense investigation.” So which is it? Is it Fe²⁺ and Fe³⁺ that is more rare? If so make this clear.

Line 124: I suspect that on this line the authors have, in a round-about way, defined what they mean by sweet and anoxic. If so, define the terms in the intro and stick with them for the remainder of the paper.

Line 126: This paragraph should end at “...natural systems.”

Speculating about “early earth”, and mentioning the current fad of CO₂ capture, are not necessary to justify this work. The work is a significant addition to the knowledge base in this field and this fact alone serves to sufficiently motivate your scientific inquiries.

Line 148: What is the notation “ca.” used for? Please clarify.

Line 149: Why use the shorthand notation for initial supersaturation if you continue to state it in words and notational form?

Line 155: Use words here “greater than 3 μ m.”

Line 156: Define the acronyms XRD and SEM at this point.

Line 168: Table 1. Here the notation ca. refers to what? Also in the second column, is this the initial concentration of solute in solution? It seems to be mixed with values for initial relative supersaturation.

Line 177: The sentence beginning on line 177 and ending on line 182 needs revision. Perhaps: “ By contrast, upon increasing the S_{CaCO₃} to ca. 10 000 (corresponding to a calcium chloride concentration of 8 mg mL⁻¹), needle shaped aragonite crystals begin to appear in one hour, alongside flower-shaped vaterite crystals (Figure 2 D);...”

Line 183: “line” ? Should possibly replace with the word “limit” ?

Line 197: I assume that the intensity units “a.u.” are clear for those that work in the X-ray diffraction field. However, a brief explanation of “a.u.” would I think benefit the wider audience.

Line 212: Write as “...less than 15 minutes.”

Line 219: Figure 3. Third sentence in caption should read “...calcite crystals and...”

Line 253: Revise this sentence as: “Rhombohedral calcite emerges as the only polymorph after the very rapid onset of crystallisation in less than 15 minutes (Figure 5 A).”

I assume the authors mean to refer to Figure 5A here, not 5C.

This situation with Ba²⁺ is intriguing. Recently, after studying the kinetics of polymorphs during the crystallization of Boc-diphenylalanine, one group found that the relative magnitudes of the

rate constants for each stage would be used to predict whether or not an Ostwald's Stage process would be observed.

[1] D.A. Barlow, Buddhi Pantha, "Kinetic Model for Ostwald's Rule of Stages with Applications to Boc-diphenylalanine Self-Assembly", *International Journal of Chemical Kinetics*, <https://doi.org/10.1002/kin.21539> (2021).

This work predicts that for stages to be observable, the rate constant for the first stage must be dominant. If not, the first stage, or stages, form but only weakly and the solute quickly passes to the final form. In the work reviewed here, Figure 5A seems to confirm this. That is, for the XRD data at the earliest time, a weak vaterite peak can be seen at $2\theta = 50^\circ$. This event is kinetically visualized by Fig. 6 in Ref. [1].

Line 257: Again, do the authors mean to refer to Figures 5B and 5C not 5A?

Line 303: Use the word "...assemble..."

Line 348: Should read: "...during a reasonable time frame such as less than 48 hours;"

Line 349: Replace all of the references to time period as $t <$ etc. with the actual words.

Line 369: Again, interestingly, in light of the results from [1] mentioned above, the paragraph starting on line 369 seems to indicate that the rate constant for formation of the first Ostwald stage is a non-linear function of the initial supersaturation.

Line 396: Should read "...ORS, crystals..."

Line 406: should read "...as this result was not observed in this study."

Line 409: Define the acronym FTIR here.

Line 416: Should read: "...show that this effect decreases..."

Line 442: Should read "However this explanation is not consistent with other reports from the literature which suggest that the presence of Fe^{2+} and Ni^{2+} ions suppress the formation of calcite altogether,..."

Line 453: Replace "evidences" with "evidence"

Line 480: The partial sentence after the semicolon needs revision or removal.

Line 486: "...in the oil and gas industry..."

Line 499: The authors capitalize "Particle Engineering" so it must be important – include a reference.

Line 502 and 504: Double check that the use of the apostrophe for "Gibbs'" is correct.

Line 507: Isn't the German word "Körper" part of the title for this famous paper?

Line 567: To be consistent, replace "Gibbs J.W." with "J.W. Gibbs"

Decision letter (RSPA-2021-0601.R0)

17-Nov-2021

Dear Dr Hadjittofis

The Editor of Proceedings A has now received comments from referees on the above paper and would like you to revise it in accordance with their suggestions which can be found below (not including confidential reports to the Editor).

Please submit a copy of your revised paper within four weeks - if we do not hear from you within this time then it will be assumed that the paper has been withdrawn. In exceptional circumstances, extensions may be possible if agreed with the Editorial Office in advance.

Please note that it is the editorial policy of Proceedings A to offer authors one round of revision in which to address changes requested by referees. If the revisions are not considered satisfactory by the Editor, then the paper will be rejected, and not considered further for publication by the journal. In the event that the author chooses not to address a referee's comments, and no scientific justification is included in their cover letter for this omission, it is at the discretion of the Editor whether to continue considering the manuscript.

To revise your manuscript, log into <http://mc.manuscriptcentral.com/prsa> and enter your Author Centre, where you will find your manuscript title listed under "Manuscripts with Decisions." Under "Actions," click on "Create a Revision." Your manuscript number has been appended to denote a revision.

You will be unable to make your revisions on the originally submitted version of the manuscript. Instead, revise your manuscript and upload a new version through your Author Centre.

When submitting your revised manuscript, you will be able to respond to the comments made by the referee(s) and upload a file "Response to Referees" in Step 1: "View and Respond to Decision Letter". Please provide a point-by-point response to the comments raised by the reviewers and the editor(s). A thorough response to these points will help us to assess your revision quickly. You can also upload a 'tracked changes' version either as part of the 'Response to reviews' or as a 'Main document'.

IMPORTANT: Your original files are available to you when you upload your revised manuscript. Please delete any unnecessary previous files before uploading your revised version.

When revising your paper please ensure that it remains under 28 pages long. In addition, any pages over 20 will be subject to a charge (£150 + VAT (where applicable) per page). Your paper has been ESTIMATED to be 28 pages.

Open Access

You are invited to opt for open access, our author pays publishing model. Payment of open access fees will enable your article to be made freely available via the Royal Society website as soon as it is ready for publication. For more information about open access please visit <https://royalsociety.org/journals/authors/open-access/>. The open access fee for this journal is £1700/\$2380/€2040 per article. VAT will be charged where applicable. Please note that if the corresponding author is at an institution that is part of a Read and Publishing deal you are required to select this option. See <https://royalsociety.org/journals/librarians/purchasing/read-and-publish/read-publish-agreements/> for further details.

Once again, thank you for submitting your manuscript to Proc. R. Soc. A and I look forward to receiving your revision. If you have any questions at all, please do not hesitate to get in touch.

Yours sincerely

Raminder Shergill

proceedingsa@royalsociety.org

on behalf of

Dr Andy Sutherland

Board Member

Proceedings A

Reviewer(s)' Comments to Author:

Referee: 1

Comments to the Author(s)

This manuscript describes the process of crystallization of CaCO_3 at high supersaturation conditions in the absence as well as in the presence of different inorganic additives, such as Li^+ , Fe^{2+} , Ba^{2+} , Ni^{2+} , Fe^{3+} cations. Within this research Authors have explored the subsequent formation of vaterite, aragonite, and calcite, exhibiting different crystal habits and studied the relation between the polymorph stability and its morphology.

Among the most important results, Authors are highlighting two observations. The first one is related to the control over the nucleation rate of CaCO_3 from its pure (not contaminated) solution. Authors show how the changes in the concentration influence the formation and stability of all three above-mentioned polymorphic forms. At low concentrations, they observed sequence of transformation, from the least stable (vaterite) with the snowflake grains, through needled-shaped aragonite, to the most stable rhombohedral calcite, whereas at increased concentration the calcite has been formed with a much faster rate. The second observation is connected to formation of the star-shaped calcite crystals in presence of cations, which can form trigonal carbonates, such as Fe^{2+} and Ni^{2+} . This is presented as a complication to the Oswald rules of Stages, as the process of crystallization is directed not only to the most stable polymorph but also toward the most stable crystal habit of certain polymorphic form. Unfortunately, these findings are overlapping with previously reported by these same Authors in the article published only few months ago (Campbell et al. 2021 Proceedings A) which strongly influences the novelty of current research.

Despite the fact, that Authors has provided sufficient techniques and adequate data to support their observations and the research done within this work is of good and general interest, in my opinion, the manuscript in its current form doesn't meet novelty requirements for Proceedings A. Especially, that the connection between the surface energy of the crystals grains and formation of the star-shaped crystals in the presence of Ni^{2+} was reported in their previous article. I am strongly convinced that this problem can be resolved, by addition of a clear and exhaustive fragment where the novelty of the current report would be expressed in the context of the previous work.

Minor issues:

in Experimental Section:

1. What was the pressure of CO_2 gas used in the experiment?
2. The source of X-ray radiation should be given.
3. How was the pH of the solution been estimated?

Results:

1. In Figures 1a and b there are some unassigned reflections (at ca. 30° , 42°), can Authors indicate their possible origin?
2. Please do replace all 'diffraction peaks' with reflections.

3. Authors are using the results of their FTIR experiments to support their findings (for example line 409 p 27, l. 451 p. 29) but no spectra or information about the instrument are provided. Unfortunately, that seems like some copy/paste leftover from Authors' previous work, where this technique was employed.

Referee: 2

Comments to the Author(s)

In this manuscript the authors report the result of an interesting collection of experiments on supersaturated CaCO_3 solutions. They clearly show the presence of Ostwald's rule of stages in the growth process and how the presence of various metal cations affect the outcome of the growth process from solution. This paper represent a significant addition to the knowledge base in this field and should be published given the minor corrections listed below.

In the abstract, line 28. Define the notation used to indicate a supersaturated condition in the intro. It is better here to state specifically: " for high relative initial supersaturations such as 2500,...". Additionally, since there are several ways of defining "supersaturation" perhaps this should be defined in the intro section as well.

Also on line 28: Define the terms "sweet" and "anoxic" later in the intro. Here be specific: "Oxygen rich solutions and oxygen depleted conditions..." Here I am speculating this is what is meant by the authors as sweet and anoxic.

Line 64: One might here, at the end of the first sentence, give the definition for initial supersaturation used in this paper and introduce the shorthand notation used by the authors: S ;

Line 91: This is a good place to define the terms "sweet" and "anoxic".

Line 101: Here the authors belatedly define the term "sweet" as used in this report. Define it earlier in the intro and remove the phrase in parenthesis here.

Line 108: The authors define the shorthand notation for supersaturated CaCO_3 in a brief fashion here. Define in a formal, and complete way earlier in the intro.

Line 112: The sentence that begins on this line ,where the authors make claims about the behavior of "calcite forming systems" upon addition of certain cations could use a reference.

Line 116: The sentence beginning with "Limited additional.." should be revised. Perhaps use: "To our knowledge, there has been a rather small number of studies on the role of additives..." Further, this is a confusing statement since the authors state on line 106 that "The role of cationic additives in the crystallisation of CaCO_3 has been a subject of intense investigation." So which is it? Is it Fe^{2+} and Fe^{3+} ; that is more rare? If so make this clear.

Line 124: I suspect that on this line the authors have, in a round-about way, defined what they mean by sweet and anoxic. If so, define the terms in the intro and stick with them for the remainder of the paper.

Line 126: This paragraph should end at "...natural systems."

Speculating about “early earth”, and mentioning the current fad of CO₂ capture, are not necessary to justify this work. The work is a significant addition to the knowledge base in this field and this fact alone serves to sufficiently motivate your scientific inquiries.

Line 148: What is the notation “ca.” used for? Please clarify.

Line 149: Why use the shorthand notation for initial supersaturation if you continue to state it in words and notational form?

Line 155: Use words here “greater than 3 μ m.”

Line 156: Define the acronyms XRD and SEM at this point.

Line 168: Table 1. Here the notation ca. refers to what? Also in the second column, is this the initial concentration of solute in solution? It seems to be mixed with values for initial relative supersaturation.

Line 177: The sentence beginning on line 177 and ending on line 182 needs revision. Perhaps: “By contrast, upon increasing the S_{CaCO₃} to ca. 10 000 (corresponding to a calcium chloride concentration of 8 mg mL⁻¹), needle shaped aragonite crystals begin to appear in one hour, alongside flower-shaped vaterite crystals (Figure 2 D);...”

Line 183: “line” ? Should possibly replace with the word “limit” ?

Line 197: I assume that the intensity units “a.u.” are clear for those that work in the X-ray diffraction field. However, a brief explanation of “a.u.” would I think benefit the wider audience.

Line 212: Write as “...less than 15 minutes.”

Line 219: Figure 3. Third sentence in caption should read “...calcite crystals and...”

Line 253: Revise this sentence as: “Rhombohedral calcite emerges as the only polymorph after the very rapid onset of crystallisation in less than 15 minutes (Figure 5 A).”

I assume the authors mean to refer to Figure 5A here, not 5C.

This situation with Ba²⁺ is intriguing. Recently, after studying the kinetics of polymorphs during the crystallization of Boc-diphenylalanine, one group found that the relative magnitudes of the rate constants for each stage would be used to predict whether or not an Ostwald’s Stage process would be observed.

[1] D.A. Barlow, Buddhi Pantha, “Kinetic Model for Ostwald’s Rule of Stages with Applications to Boc-diphenylalanine Self-Assembly”, International Journal of Chemical Kinetics, <https://doi.org/10.1002/kin.215392021> (2021).

This work predicts that for stages to be observable, the rate constant for the first stage must be dominant. If not, the first stage, or stages, form but only weakly and the solute quickly passes to the final form. In the work reviewed here, Figure 5A seems to confirm this. That is, for the XRD data at the earliest time, a weak vaterite peak can be seen at 2 θ 50^o. This event is kinetically visualized by Fig. 6 in Ref. [1].

Line 257: Again, do the authors mean to refer to Figures 5B and 5C not 5A?

Line 303: Use the word “...assemble...”

Line 348: Should read: “...during a reasonable time frame such as less than 48 hours;”

Line 349: Replace all of the references to time period as t< etc. with the actual words.

Line 369: Again, interestingly, in lite of the results from [1] mentioned above, the paragraph starting on line 369 seems to indicate that the rate constant for formation of the first Ostwald stage is a non-linear function of the initial supersaturation.

Line 396: Should read “...ORS, crystals...”

Line 406: should read “...as this result was not observed in this study.”

Line 409: Define the acronym FTIR here.

Line 416: Should read: “...show that this effect decreases...”

Line 442: Should read “However this explanation is not consistent with other reports from the literature which suggest that the presence of Fe²⁺ and Ni²⁺ ions suppress the formation of calcite altogether...”

Line 453: Replace “evidences” with “evidence”

Line 480: The partial sentence after the semicolon needs revision or removal.

Line 486: “..in the oil and gas industry...”

Line 499: The authors capitalize “Particle Engineering” so it must be important – include a reference.

Line 502 and 504: Double check that the use of the apostrophe for “Gibbs’ ” is correct.

Line 507: Isn’t the German word “Korper” part of the title for this famous paper?

Line 567: To be consistent, replace “Gibbs J.W.” with “J.W. Gibbs”

Board Member:

Comments to Author(s):

Please can you address the concerns of the two referees, especially the one regarding the novelty of the work described in the current manuscript given your recent publications in this area. Many thanks.

Author's Response to Decision Letter for (RSPA-2021-0601.R0)

See Appendix A.

Decision letter (RSPA-2021-0601.R1)

05-Jan-2022

Dear Dr Hadjittofis

I am pleased to inform you that your manuscript entitled "Exploring the role of crystal habit in the Ostwald rule of stages" has been accepted in its final form for publication in Proceedings A.

Our Production Office will be in contact with you in due course. You can expect to receive a proof of your article soon. Please contact the office to let us know if you are likely to be away from e-mail in the near future. If you do not notify us and comments are not received within 5 days of sending the proof, we may publish the paper as it stands.

As a reminder, you have provided the following 'Data accessibility statement' (if applicable). Please remember to make any data sets live prior to publication, and update any links as needed when you receive a proof to check. It is good practice to also add data sets to your reference list. Statement (if applicable):

Under the terms of our licence to publish you may post the author generated postprint (ie. your accepted version not the final typeset version) of your manuscript at any time and this can be made freely available. Postprints can be deposited on a personal or institutional website, or a recognised server/repository. Please note however, that the reporting of postprints is subject to a media embargo, and that the status the manuscript should be made clear. Upon publication of the definitive version on the publisher's site, full details and a link should be added.

You can cite the article in advance of publication using its DOI. The DOI will take the form: 10.1098/rspa.XXXX.YYYY, where XXXX and YYYY are the last 8 digits of your manuscript number (eg. if your manuscript number is RSPA-2017-1234 the DOI would be 10.1098/rspa.2017.1234).

For tips on promoting your accepted paper see our blog post:
<https://royalsociety.org/blog/2020/07/promoting-your-latest-paper-and-tracking-your-results/>

On behalf of the Editor of Proceedings A, we look forward to your continued contributions to the Journal.

Sincerely,
Raminder Shergill
proceedingsa@royalsociety.org

on behalf of
Dr Andy Sutherland
Board Member
Proceedings A

Appendix A

The authors thank the reviewers and the editorial team for their effort.

This document summarises the authors' reply to the reviewer's comments.

For Reviewer 1, an extended part of the reply is dedicated to address the novelty concerns raised.

Reviewer 1

We, the authors, are extremely appreciative provided by the reviewer, particularly for the comments; additionally, the authors are grateful to be given the opportunity to elaborate on the novelty of this work. The reviewer is correct that this manuscript is intimately connected to the work presented in the recent publication. We are pleased that the reviewer has seen this, to understand that the two papers (published and this manuscript) are part of a large investigation. In working to write up this work a wide range of options were considered where every effort to make a single manuscript was cumbersome, and, frankly speaking, difficult to read. As such, after trying it was decided to divide the work into two manuscripts. However, it did continue to be a 'chicken and egg' problem as to which to submit first.

The published paper provides the observations; these results motivated the significant additional experiments used to develop understanding and subsequently propose a mechanism for the phenomena observed described in this manuscript.

In the published paper, the role of surface energy and particle anisotropy on solid state properties was discussed, on the ground of a certain number of samples and measurements. This complete manuscript included a clear research hypothesis, experimental work, and subsequent discussion and conclusions; it finely illustrated how particles' surface energy anisotropy can potentially impact their apparent solubility, the same way that polymorphism does. Thus, the interplay between surface and bulk properties was established. The methodology used for the preparation of the star-shaped particles was outlined for only one specific additive. Critically, mechanistic understanding of the roots and the causes of the particular Ostwald rule of stages (ORS) behaviour was not scrutinized, therefore leaving a gap in knowledge to explain these observations.

In this manuscript the work employs additional experiments to explain the phenomena observed in the published paper. A detailed experimental workflow was deployed to decipher the role of supersaturation, cationic additive type and size, and additive concentration on the cascade of ORS transformations. Through the detailed experimental matrix, deployed in this paper, a number of important research questions are answered, resulting in a significant leap in understanding (that is not even considered in the published paper). It is demonstrated that at the set of conditions of interest, the ORS is a critical mechanism, determining the formation of CaCO₃ particles. Conclusions uniquely drawn in this manuscript include:

- It is established that the type and the size of cationic additives impact significantly the supersaturation degree and consequentially the extent of the ORS cascade of transformations.
- Most importantly it is demonstrated that specific cations, with the ability to form carbonates with the same crystal structure as calcite, can facilitate the early nucleation of calcite, at conditions where vaterite should emerge.
- Following this, it is shown that under these supersaturation conditions, the calcite crystals grow to the star-shaped crystal habit featured in the first paper.
- Finally, thanks to the meticulous review effort, our results were found to link very well with a novel mathematical description of the ORS.

Overall, these items constitute important findings advancing the community's knowledge in the intriguing field of crystallisation of carbonates under high supersaturations, in the absence and presence of cationic additives. The findings provided can shed light on the formation of minerals in eras and locations where sweet and anoxic conditions are/were dominant (this includes natural and artificial systems). Most importantly these findings provide a roadmap for the development of strategies towards the production of high surface energy particles, which can be very beneficial in fields like drug product development.

Minor issues:

in Experimental Section:

1. What was the pressure of CO₂ gas used in the experiment?

The pressure of the CO₂ gas was kept at 0.5 bar. This point was amended in the document.

2. The source of X-ray radiation should be given.

The XRPD was using Cu-K α radiation. This point was amended in the document.

3. How was the pH of the solution been estimated?

The pH was not estimated, it was measured using a pH meter from Mettler Toledo. The points was amended in the document

Results:

1. In Figures 1a and b there are some unassigned reflections (at ca. 30°, 42°), can Authors indicate their possible origin?

The latter has been identified as an aragonite peak, while for the former the authors believe there is some ambiguity regarding its possible origin.

2. Please do replace all 'diffraction peaks' with reflections.

This point was amended in the document

3. Authors are using the results of their FTIR experiments to support their findings (for example line 409 p 27, l. 451 p. 29) but no spectra or information about the instrument are provided. Unfortunately, that seems like some copy/paste leftover from Authors' previous work, where this technique was employed.

The authors have not included the FTIR spectrum as they are already in a previous manuscript. Therefore, a reference to that paper has been included in the review

Referee: 2

Comments to the Author(s)

In this manuscript the authors report the result of an interesting collection of experiments on supersaturated CaCO₃ solutions. They clearly show the presence of Ostwald's rule of stages in the growth process and how the presence of various metal cations affect the outcome of the growth process from solution. This paper represent a significant addition to the knowledge base in this field and should be published given the minor corrections listed below.

The authors thank the reviewer for his very meticulous review effort. Most importantly, they would like to highlight its contribution in elevating the content and the conclusions of this work by providing a recent piece of work, very relevant to the topic.

In the abstract, line 28. Define the notation used to indicate a supersaturated condition in the intro. It is better here to state specifically: “ for high relative initial supersaturations such as 2500,...”. Additionally, since there are several ways of defining “supersaturation” perhaps this should be defined in the intro section as well.

The authors thank the reviewer for this comment. This comment, along with others, helped us to improve the consistency of the manuscript. The comment was amended in the document.

Also on line 28: Define the terms “sweet” and “anoxic” later in the intro. Here be specific: “Oxygen rich solutions and oxygen depleted conditions...” Here I am speculating this is what is meant by the authors as sweet and anoxic.

This was also amended in the document.

Line 64: One might here, at the end of the first sentence, give the definition for initial supersaturation used in this paper and introduce the shorthand notation used by the authors: S_{CaCO_3}

The authors introduced this notation earlier on.

Line 91: This is a good place to define the terms “sweet” and “anoxic”.

This was also amended earlier on in the reviewed document.

Line 101: Here the authors belatedly define the term “sweet” as used in this report. Define it earlier in the intro and remove the phrase in parenthesis here.

This was amended in the document.

Line 108: The authors define the shorthand notation for supersaturated $CaCO_3$ in a brief fashion here. Define in a formal, and complete way earlier in the intro.

This was amended in the document.

Line 112: The sentence that begins on this line ,where the authors make claims about the behavior of “calcite forming systems” upon addition of certain cations could use a reference.

This was amended in the document.

Line 116: The sentence beginning with “Limited additional..” should be revised. Perhaps use: “To our knowledge, there has been a rather small number of studies on the role of additives...”

Further, this is a confusing statement since the authors state on line 106 that “The role of cationic additives in the crystallisation of $CaCO_3$ has been a subject of intense investigation.” So which is it? Is it Fe^{2+} and Fe^{3+} that is more rare? If so make this clear.

Indeed, the initial form of this sentence was creating ambiguity regarding the extent of the available literature. Even if there are some studies on the role of additives, not many though, only a fraction of them are dealing with Fe^{2+} and Fe^{3+} . The authors amended this in the document.

Line 124: I suspect that on this line the authors have, in a round-about way, defined what they mean by sweet and anoxic. If so, define the terms in the intro and stick with them for the remainder of the paper.
The authors have fixed all the cases of terminology inconsistencies identified.

Line 126: This paragraph should end at "...natural systems."
Speculating about "early earth", and mentioning the current fad of CO₂ capture, are not necessary to justify this work. The work is a significant addition to the knowledge base in this field and this fact alone serves to sufficiently motivate your scientific inquiries.
The authors thank the reviewer for this very positive comment. The point was amended in the document.

Line 148: What is the notation "ca." used for? Please clarify.
The annotation ca. stems from the Roman circa meaning "about". Therefore, it should be written in italics. The authors have amended this throughout the document.

Line 149: Why use the shorthand notation for initial supersaturation if you continue to state it in words and notational form?
This is another inconsistency the authors amended.

Line 155: Use words here "greater than 3 μ m."
The authors have removed any > < signs from the document.

Line 156: Define the acronyms XRD and SEM at this point.
The authors have done so.

Line 168: Table 1. Here the notation ca. refers to what? Also in the second column, is this the initial concentration of solute in solution? It seems to be mixed with values for initial relative supersaturation.
The table features, below the values for concentration, the corresponding values of supersaturation in systems do not containing other cations. A note has been added in the document to explain.

Line 177: The sentence beginning on line 177 and ending on line 182 needs revision. Perhaps: "By contrast, upon increasing the S_{CaCO₃} to ca. 10 000 (corresponding to a calcium chloride concentration of 8 mg mL⁻¹), needle shaped aragonite crystals begin to appear in one hour, alongside flower-shaped vaterite crystals (Figure 2 D);..."
The authors amended this sentence in the reviewed document.

Line 183: "line" ? Should possibly replace with the word "limit" ?
The word limit, indeed, provides a better description. The authors amended this in the document.

Line 197: I assume that the intensity units "a.u." are clear for those that work in the X-ray diffraction field. However, a brief explanation of "a.u." would I think benefit the wider audience.
The authors added an explanation on this inside the document.

Line 212: Write as "...less than 15 minutes."

The authors amended this sentence in the reviewed document.

Line 219: Figure 3. Third sentence in caption should read "...calcite crystals and..."

The authors amended this sentence in the reviewed document.

Line 253: Revise this sentence as: "Rhombohedral calcite emerges as the only polymorph after the very rapid onset of crystallisation in less than 15 minutes (Figure 5 A)."

I assume the authors mean to refer to Figure 5A here, not 5C.

The authors amended this sentence in the reviewed document, as indeed it was not making sense as it was written initially.

This situation with Ba^{2+} is intriguing. Recently, after studying the kinetics of polymorphs during the crystallization of Boc-diphenylalanine, one group found that the relative magnitudes of the rate constants for each stage would be used to predict whether or not an Ostwald's Stage process would be observed.

[1] D.A. Barlow, Buddhi Pantha, "Kinetic Model for Ostwald's Rule of Stages with Applications to Boc-diphenylalanine Self-Assembly", International Journal of Chemical Kinetics, <https://doi.org/10.1002/kin.215392021> (2021).

This work predicts that for stages to be observable, the rate constant for the first stage must be dominant. If not, the first stage, or stages, form but only weakly and the solute quickly passes to the final form. In the work reviewed here, Figure 5A seems to confirm this. That is, for the XRD data at the earliest time, a weak vaterite peak can be seen at 2θ 50° . This event is kinetically visualized by Fig. 6 in Ref. [1].

This paper really facilitates the discussion of the results, not only regarding the appearance of calcite in the presence of Ba^{2+} and in Figure 5A, but also regarding the appearance of the star-shaped calcite crystals.

The authors have used the result of this paper as the launchpad to explain results, at different parts of the discussion section.

Line 257: Again, do the authors mean to refer to Figures 5B and 5C not 5A?

The authors amended this in the reviewed document.

Line 303: Use the word "...assemble..."

The authors amended this in the reviewed document.

Line 348: Should read: "...during a reasonable time frame such as less than 48 hours;"

The authors amended this in the reviewed document.

Line 349: Replace all of the references to time period as $t <$ etc. with the actual words.

The authors amended this in the reviewed document.

Line 369: Again, interestingly, in light of the results from [1] mentioned above, the paragraph starting on line 369 seems to indicate that the rate constant for formation of the first Ostwald stage is a non-linear function of the initial supersaturation.

The authors amended this in the reviewed document.

Line 396: Should read "...ORS, crystals..."

The authors amended this in the reviewed document.

Line 406: should read "...as this result was not observed in this study."

The authors amended this in the reviewed document.

Line 409: Define the acronym FTIR here.

The authors amended this in the reviewed document.

Line 416: Should read: "...show that this effect decreases..."

The authors amended this in the reviewed document.

Line 442: Should read "However this explanation is not consistent with other reports from the literature which suggest that the presence of Fe^{2+} and Ni^{2+} ions suppress the formation of calcite altogether,..."

The authors amended this sentence in the reviewed document.

Line 453: Replace "evidences" with "evidence"

The authors amended this in the reviewed document.

Line 480: The partial sentence after the semicolon needs revision or removal.

The authors amended this in the reviewed document.

Line 486: "...in the oil and gas industry..."

The authors amended this in the reviewed document.

Line 499: The authors capitalize "Particle Engineering" so it must be important—include a reference.

A reference was added.

Line 502 and 504: Double check that the use of the apostrophe for "Gibbs' " is correct.

As part of the review process, this part was erased

Line 507: Isn't the German word "Körper" part of the title for this famous paper?

The authors amended this sentence in the reviewed document.

Line 567: To be consistent, replace "Gibbs J.W." with "J.W. Gibbs"
The authors amended this in the reviewed document.